# Heteroaryl-Ethylenes as New Effective Agents for High Priority Gram-Positive and Gram-Negative Bacterial Clinical Isolates

**DOI:** 10.3390/antibiotics11060767

**Published:** 2022-06-03

**Authors:** Dalida Angela Bivona, Alessia Mirabile, Carmelo Bonomo, Paolo Giuseppe Bonacci, Stefano Stracquadanio, Andrea Marino, Floriana Campanile, Carmela Bonaccorso, Cosimo Gianluca Fortuna, Stefania Stefani, Nicolò Musso, Dafne Bongiorno

**Affiliations:** 1Section of Microbiology, Department of Biomedical and Biotechnological Sciences (BIOMETEC), University of Catania, 95123 Catania, Italy; dalidabivona@gmail.com (D.A.B.); alessiamirabile93@gmail.com (A.M.); carmelo.bonomo@phd.unict.it (C.B.); paolo.bonacci@unict.it (P.G.B.); s.stracquadanio@unict.it (S.S.); andreamarino9103@gmail.com (A.M.); f.campanile@unict.it (F.C.); stefania.stefani@unict.it (S.S.); nmusso@unict.it (N.M.); 2Department of Chemical Sciences, University of Catania, 95125 Catania, Italy; bonaccorsoc@gmail.com (C.B.); cg.fortuna@unict.it (C.G.F.)

**Keywords:** ESKAPE, heteroaryl-ethylene, clinical strains, antimicrobial activity

## Abstract

The World Health Organization has identified antimicrobial resistance as a public health emergency and developed a global priority pathogens list of antibiotic-resistant bacteria that can be summarized in the acronym ESKAPE (*Enterococcus faecium*, *Staphylococcus aureus*, *Klebsiella pneumoniae*, *Acinetobacter baumannii*, *Pseudomonas aeruginosa* and *Enterobacterales* species), reminding us of their ability to escape the effect of antibacterial drugs. We previously tested new heteroaryl-ethylene compounds in order to define their spectrum of activity and antibacterial capability. Now, we focus our attention on PB4, a compound with promising MIC and MBC values in all conditions tested. In the present study, we evaluate the activity of PB4 on selected samples of ESKAPE isolates from nosocomial infections: 14 *S. aureus*, 6 *E. faecalis*, 7 *E. faecium*, 12 *E. coli* and 14 *A. baumannii*. Furthermore, an ATCC control strain was selected for all species tested. The MIC tests were performed according to the standard method. The PB4 MIC values were within very low ranges regardless of bacterial species and resistance profiles: from 0.12 to 2 mg/L for *S. aureus*, *E. faecalis*, *E. faecium* and *A. baumannii*. For *E. coli*, the MIC values obtained were slightly higher (4–64 mg/L) but still promising. The PB4 heteroaryl-ethylenic compound was able to counteract the bacterial growth of both high-priority Gram-positive and Gram-negative clinical strains. Our study contributes to the search for new molecules that can fight bacterial infections, in particular those caused by MDR bacteria in hospitals. In the future, it would be interesting to evaluate the activity of PB4 in animal models to test for its toxicity.

## 1. Introduction

An important development of modern medicine is the efficient treatment of bacterial infections with antibiotics but, over the years, bacteria have found ways to resist the action of antimicrobial drugs [1,2].

Antimicrobial resistance (AMR) is a crucial global concern due to the increased and irresponsible use of antibiotics promoting the selection and spread of antibiotic-resistant pathogens responsible for difficult-to-treat infections, especially in Intensive Care Units (ICUs) [3].

Infections caused by multi-drug resistant (MDR) pathogens can lead to additional complications such as prolonged hospital stays and protracted treatment with last-line antibiotics, which increases the selection of resistant microbiota and results in higher healthcare costs. Bacterial resistance can be both intrinsic and acquired, the latter possibly being greatly enhanced by antimicrobial drugs exposure [4].

In 2017, the World Health Organization (WHO) created an acronym, ESKAPE, to represent the *Enterococcus faecium*, *Staphylococcus aureus*, *Klebsiella pneumoniae*, *Acinetobacter baumannii*, *Pseudomonas aeruginosa*, and Enterobacterales species [5]; to date, we are still facing a worldwide prevalence of MDR pathogens.

Gram-positive bacteria (GPB) have developed particular resistance profiles, such as Methicillin-Resistant *Staphylococcus aureus* (MRSA) [6] or Vancomycin-Resistant enterococci (VRE) [7] and are often related to nosocomial infections. Both these pathogens are responsible for increased morbidity and mortality due to the ineffectiveness of “last resort” antibiotics [8,9].

MRSA strains are characterized by a mobile genetic element carrying the methicillin-resistance gene (*mec*A), called Staphylococcal Chromosomal Cassette (SCC*mec*), usually type I, II or III [10,11]. Clinically, the acquisition of resistance to multiple antibiotic classes by *S. aureus* complicates treatment [12]. *E. faecalis* and *E. faecium* are responsible for the majority of human infections and related to the presence of bladder catheter, and/or neutropenia [13,14]. Nosocomial infections with enterococci are often associated with peculiar antimicrobial resistance profiles exhibiting a high level of intrinsic resistance to penicillins, cephalosporins, aminoglycosides and carbapenems [15]. As shown above, the WHO list mainly consists of Gram-negative bacteria (GNB). *Enterobacterales* resistance to third generation cephalosporins and carbapenems is increasing worldwide and is now above 10% and 2–7%, respectively [4]. This is due to the spread of Extended Spectrum Beta-Lactamases (ESBLs) (CTX-M, TEM, SHV) and carbapenemases (VIM, IMP, NDM, KPC and OXA) [16]. 

*E. coli* and *K. pneumoniae* are ESBL producers; compared with non-ESBLs, they express more TEM1, TEM2 and SHV capable of hydrolyzing narrow spectrum cephalosporins, carbapenems and monobactams. The administration of piperacillin/tazobactam, a treatment that alternates a β-lactam and β-lactamase inhibitor, has been considered as a carbapenem-sparing regimen for ESBL infections, although the global trend of AmpC β-lactamase-producing bacteria should be carefully monitored [17]. *A. baumannii* is one of the most successful pathogens in causing nosocomial infections, and carbapenem-resistant *A. baumannii* (CRAB) is often isolated in hospital settings. Current antimicrobials for CRAB (i.e., polymyxins, tigecycline, and sometimes aminoglycosides) are far from perfect therapeutic options due to their pharmacokinetic properties and increasing resistance rates [18,19].

In such a critical situation, the introduction in the clinical practice of new molecules able to counteract the continuous increase of multidrug-resistant pathogens is unquestionably a current priority [20]. In fact, while the constant introduction of new drugs has long made it possible to bypass the issue of antibiotic resistance, the current lack of molecules effective against MDR pathogens is becoming of growing concern [21]. 

In a previous study by Bongiorno et al. [22], the spectrum of ability of eight heteroaryl-ethylene compounds, called PBn, as antimicrobial agents capable of inhibiting the proliferation of a selected sample of Gram-positive and Gram-negative ATCC strains was tested by MIC (minimal inhibition concentration) and MBC (minimal bactericidal concentration) assays. Furthermore, the presence of an inoculum effect was assessed at scalar inoculum concentrations, and the cytotoxicity of the new molecules on colorectal adenocarcinoma cancer cells (CaCo_2_) was analyzed [22]. Our preliminary results were highly encouraging and pave the way for further investigations of heteroaryl-ethylenes as antimicrobial agents in the treatment of several Gram-positive and Gram-negative bacterial infections, especially those caused by MRSA, VRE, ESBL and *A. baumannii*. 

In the present study, we have focused our attention on PB4 (Figure 1) which emerged previously as the most promising PBn heteroaryl-ethylene compound with low MIC and MBC values in all conditions tested [22].

PB4 activity was evaluated on selected samples of MRSA, VRE, ESBL and CRAB freshly isolated from nosocomial infections. The workflow is shown in Figure 2. 

## 2. Results

In this study, we selected Gram-positive and Gram-negative clinical strains with different antibiotic resistance profiles, including 14 *S. aureus*, 6 *E. faecalis*, 7 *E. faecium*, 14 *A. baumannii*, and 12 *E. coli*, aiming to understand the potency and antimicrobial capacity of PB4. Furthermore, an ATCC control strain was selected for all species tested. 

### 2.1. Gram-Positive Clinical Strains

PB4 was tested against both MRSA and MSSA isolates (11 and 3 strains, respectively). PB4 showed an antimicrobial activity of 0.25 mg/L MIC on *S. aureus* ATCC29213. The MIC values of PB4 were within a range of 0.12 mg/L to 0.5 mg/L for all *S. aureus* strains tested. PB4 was active on 3 MSSA and 8 MRSA at a concentration of 0.12 mg/L, proving effective at very low concentrations despite the greater aggressiveness of the control MRSA strains included in the study. Slightly higher concentrations of PB4 (0.25 mg/L and 0.5 mg/L) were sufficient to inhibit the growth of the three remaining clinical MRSA strains (see Table 1).

*Enterococcus* spp. clinical strains were selected for different antibiotic resistance profiles, including 6 *E. faecalis* (VRE, MDR, Not MDR) and 7 *E. faecium* (VRE, Linezolid Resistant (LinR), MDR, Not MDR). *E. faecalis* ATCC 29212 and ATCC 51299, with a PB4 MIC value of 0.5 mg/L, were included. The MIC values for PB4 were in a range between 0.12 mg/L and 2 mg/L when tested on *Enterococci*.

In particular, PB4, independently of their antibiotic profile, showed a MIC range of 0.25 to 0.5 mg/L for *E. faecalis* strains, and of 0.12 to 2 mg/L for *E. faecium*. The MIC values for PB4 in the *Enterococcus* spp. sample are reported in Table 2.

### 2.2. Gram-Negative Clinical Strains

Furthermore, 12 clinical strains were selected for their different antibiotic resistance profiles, in particular 3 ESBL and 9 non-ESBL *E. coli*.

*E. coli* ATCC 25922 was chosen as a control strain in order to test the activity of PB4 on this species, and the compound showed a MIC value of 1 mg/L. The MIC values for PB4 were in a range between 4 mg/L and 64 mg/L. The MIC values for PB4 were heterogeneous on *E. coli*, showing different behaviors for the three different ESBL clinical strains. Regarding non-ESBL *E. coli*, PB4 MIC values of 4 mg/L were obtained for five clinical strains, 8 mg/L for one clinical strain, and 32 mg/L for the two remaining clinical strains (see Table 3).

Regarding clinical strains, we selected 14 clinical MDR strains based on their carbapenem-resistance profiles, particularly Carbapenem Resistant *A. baumannii* (CRAB). We obtained PB4 MIC values in a range between <0.12 mg/L and 1 mg/L (see Table 4).

A previous observation showed that PB4 may be active on *A. baumannii* ATCC 17978 control strain, with an antimicrobial activity of 0.5 mg/L MIC.

A summary table (see Table 5) of the results obtained clarifies the activity of PB4. The PB4 MIC values obtained on the control strains are reported on the far left of the table as a reference.

In general, we have seen that PB4 is effective at very low ranges, and this is true for all Gram-positive and Gram-negative clinical strains tested regardless of the species and their particular antibiotic resistance profile. 

On *S. aureus* clinical strains, PB4 showed an inhibitory efficacy in a range between 0.12 and 0.5 mg/L: 11/14 clinical strains exhibited the lowest recorded MIC values of 0.12 mg/L; 1/14 had values of 0.25 mg/L; and 2/14 had values of 0.5 mg/L. A PB4 MIC value of 0.25 mg/L was measured for *S. aureus* ATCC 29213.

On *E. faecalis* clinical strains, PB4 was effective in a range between 0.25 and 0.5 mg/L: 4/6 clinical strains had the lowest recorded MIC value of 0.25 mg/L; and 2/6 strains had values of 0.5 mg/L.

On *E. faecium* clinical strains, PB4 activity was detected in a range between 0.12 and 2 mg/L: 3/7 clinical strains had low MIC values in a range of 0.12 to 0.25–0.5 mg/L; 3/7 showed an intermediate MIC value of 1 mg/L, and the last clinical strain exhibited the highest MIC value of 2 mg/L. For *E. faecalis* ATCC 29212 and *E. faecalis* ATCC 51299, a PB4 MIC value of 0.5 mg/L was measured.

On *E. coli* clinical strains, PB4 efficacy was assessed in a range between 4 and 64 mg/L: 6/12 clinical strains exhibited the lowest MIC value of 4 mg/L; 1/12 had a value of 8 mg/L; 1/12 had a value of 16 mg/L; 2/12 had a value of 32 mg/L; and only 2/12 strains had the highest MIC value of 64 mg/L. For *E. coli* ATCC 25922, a PB4 MIC value of 1 mg/L was measured.

On *A. baumannii* clinical strains, PB4 inhibition activity ranged between ≤0.12 and 1 mg/L: 4/14 clinical strains had the lowest MIC value of less than or equal to 0.12 mg/L; 6/14 had values of 0.25 mg/L; 3/14 had values of 0.5 mg/L; and 1/14 had a value of 1 mg/L. For *A. baumannii* ATCC 17978, a PB4 MIC value of 0.5 mg/L was measured.

## 3. Discussion

In 2017, the World Health Organization (WHO) published a list of antibiotic-resistant priority pathogens that can be summarized in the acronym ESKAPE (*E. faecium*, *S. aureus*, *K. pneumoniae*, *A. baumannii*, *P. aeruginosa* and *Enterobacter* spp.), reminding us of their ability to escape the effect of antibacterial drugs, which, in turn, accounts for their ability to cause serious, difficult-to-control Gram-positive and Gram-negative bacterial infections, especially in the hospital setting [23].

Thus, we focused our attention on *S. aureus* (MSSA and MRSA), VR and LinR *E. faecalis* and *E. faecium*, ESBL *E. coli*, and carbapenem resistant *A. baumannii* (CRAB). For this purpose, in a previous study we had tested newly synthesized molecules derived from the condensation of heterocyclic aromatic aldehydes: the PB compounds. These eight molecules were tested on Gram-positive and Gram-negative ATCC control strains to characterize their antimicrobial activity and determine their MIC and MCB values [22].

Based on previous results, PB4 was selected as the compound candidate to be tested on MSSA and MRSA, VRE and LinR *Enterococci*, ESBL *E. coli*, and CRAB.

Surprisingly, despite the antimicrobial resistance profile which characterizes every single clinical strain, PB4 showed a good level of activity. Indeed, the MIC values of PB4 were within very low ranges regardless of bacterial species and resistance profiles: from 0.12 to 0.5 mg/L for both MSSA and MRSA; from 0.25 to 0.5 mg/L for *E. faecalis*, and from 0.12 to 2 mg/L for *E. faecium*. 

Only for *E. coli*, including the ATCC strain, which showed a MIC value of 1 mg/L, the MIC values obtained were from 4 mg/L to 64 mg/L, slightly higher than for other microorganisms tested but still promising. Remarkably, for *A. baumannii* clinical strains, the inhibitory activity of PB4 ranged from ≤0.12 mg/L to 1 mg/L, despite their MDR, especially, their resistance to carbapenems (CRAB).

As discussed above, PB4 acts on all clinical strains, both Gram-positive and Gram-negative, in very low ranges. Furthermore, values were comparable for organisms belonging to the same species but with different antibiotic resistance profiles. For *E. coli*, we observed a slightly higher PB4 MIC range but there was no difference between the ESBL and Susceptible strains that were analyzed. Thus, the different PB4 action is due to the unique characteristics of each strain.

It is very important to highlight that several ranges of action in microorganisms belonging to different species does not indicate a poor molecule functionality. The MIC value and its action range are usually species specific [24]. To date, the PB4 target is still unknown. Probably, the molecule does not act on a site characteristic to Gram-positive or Gram-negative but on their common structure. We intend to study the toxicity, pharmacokinetics and pharmacodynamics of PB4 to confirm this surprising antimicrobial activity acting on difficult-to-treat infections. 

In this study, we observed that PB4, a heteroaryl-ethylenic compound, can counteract the bacterial growth of both high-priority Gram-positive and Gram-negative clinical strains. Our aim was to contribute to the search for new molecules that can fight bacterial infections, in particular those caused by MDR bacteria in hospitals, where serious infections, combined with impaired pathophysiology and immunity of patients, cannot but worsen their clinical picture and outcome [25].

In the future, in order to better investigate the good antimicrobial capability found in bacterial strains through in vitro studies and in prokaryotic cells, it would be interesting to test PB4 for its toxicity and antimicrobial activity using in vivo models.

This is a crucial aspect, because the molecule could fail in vivo depending on its toxicity or rate of elimination by the organism to which it is administered.

## 4. Materials and Methods

### 4.1. Characteristics of Molecule Tested in the Study 

PB4 (Pyridinium, 2,6-bis[(1E)-2-[4-(dimethylamino)phenyl]ethenyl]-1-methyl-, iodide CAS: 793694-41-4) was synthesized as previously reported [26] and characterized by ^1^H NMR, ^13^C NMR, and MS spectrometry. The purity of PB4 was confirmed through NMR and TGA analysis; the batch employed for these experiments reached a purity degree of 96%.

### 4.2. Microorganisms and Growth Conditions

Control strains of Gram-positive and Gram-negative bacteria, *E. faecalis* ATCC 29212 VSE, *E. faecalis* ATCC 51299 VRE, *S. aureus* ATCC 29213, *E. coli* ATCC 25922, *A. baumannii* ATCC 17978, were obtained from the American Type Culture Collection (Manassas, VA, USA).

In this study, fourteen *S. aureus*, six *E. faecalis*, seven *E. faecium* and fourteen *A. baumannii* clinical strains were selected for their resistance profile among the bacterial culture collections of the Medical Molecular Microbiology and Antibiotic Resistance Laboratory (MMARL) of the University of Catania.

The identification of Gram-positive strains was previously conducted using API system (Biomérieux, Craponne , France) and confirmed by sequencing the 16S gene [27,28,29].

The identification of *A. baumannii* strains was confirmed by Matrix-Assisted Laser Desorption Ionization–Time of Flight (MALDI87 TOF) mass spectrometry (Bruker Daltonics, Billerica, MA, USA) [30].

Twelve *E. coli* clinical strains were provided by the Laboratory of Microbiology and Clinical Virology, Gaspare Rodolico Hospital of Catania, Vittorio Emanuele University Hospital.

The identification of the strains was performed by Vitek2 GN (Biomérieux, France) systems. 

*S. aureus* was grown on Mannitol Salt Agar (CM0085B, Thermo Scientific^TM^ Oxoid^TM^, Basingstoke, UK), *E. faecalis* on Bile Aesculin Agar (CM0888, Thermo Scientific^TM^ Oxoid^TM^, Basingstoke, UK), *A. baumannii* and *E. coli* on MacConkey Agar (CM0007, Thermo Scientific^TM^ Oxoid^TM^, Basingstoke, UK) at 37 °C for 24 h.

### 4.3. Minimum Inhibitory Concentration (MIC) 

Microtiter plate assays were performed to determine the minimum inhibitory concentration of PB4 according to the standard method [31,32,33,34], with some modifications. 

The PB4 compound was at a concentration of 8000 mg/L in 100% DMSO (85190, Thermo Scientific^TM^ Oxoid^TM^, Basingstoke, UK). Further dilutions of the substance were prepared using Mueller Hinton II Broth [Cation-Adjusted] (CA-MHB) (212322, BD BBLTM, Franklin Lakes, NJ, USA).

The concentration range tested was between 128 mg/L and 0.25 mg/L. A 100 μL aliquot of PB4 was inoculated in 96-well microplates containing sterile CA-MHB, and serial dilutions were performed. 

Subsequently, starting from different 0.5 McFarland (10^8^ CFU/mL) bacterial suspensions, scalar dilutions of the inoculum were carried out and a concentration of 10^5^ CFU/mL was inoculated in 96-well microplates. The microplates were incubated for 18 ± 2 h at 37 °C. MIC values were determined at the lowest concentrations of the antimicrobial agent inhibiting bacterial growth. The MIC values were expressed in mg/L. The tests were repeated in duplicate by two different researchers. 

## Figures and Tables

**Figure 1 antibiotics-11-00767-f001:**
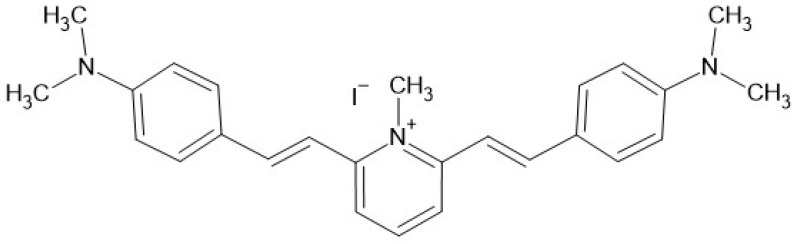
Chemical structure of PB4, the figure was created using BioRender bioinformatic tool.

**Figure 2 antibiotics-11-00767-f002:**
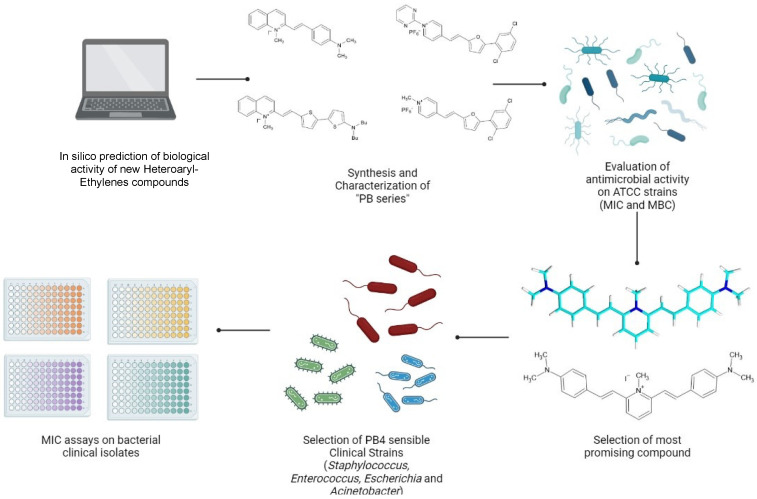
Experimental workflow, the figure was created using BioRender bioinformatic tool.

**Table 1 antibiotics-11-00767-t001:** Activity of PB4 on *S. aureus* ATCC 29213 control strain and on 14 clinical *S. aureus* strains (3 MSSA and 11 MRSA).

Antimicrobial Resistance Profile	PB4 MIC		
	Strain	mg/L	µmol/L
	*S. aureus* ATCC 29213	0.25	0.49
MSSA	1-CT	0.12	0.234
	2-CT	0.12	0.234
	3-CT	0.12	0.234
MRSA	4-CT	0.12	0.234
	5-CT	0.12	0.234
	6-CT	0.12	0.234
	7-CT	0.12	0.234
	8-CT	0.12	0.234
	9-CT	0.12	0.234
	10-CT	0.12	0.234
	11-CT	0.12	0.234
	12-CT	0.25	0.49
	13-CT	0.5	0.977
	14-CT	0.5	0.977

**Table 2 antibiotics-11-00767-t002:** Activity of PB4 on *E. faecalis* ATCC 29212 and *E. faecalis* ATCC 51299 control strains, on six *E. faecalis* and on seven *E. faecium* clinical strains (VRE, LinR, MDR, not MDR).

Antimicrobial Resistance Profile		PB4 MIC		
		Strain	mg/L	µmol/L
		*E. faecalis* ATCC 29212	0.5	0.977
		*E. faecalis* ATCC 51299	0.5	0.977
*E. faecalis*	VRE	15-CT	0.25	0.488
		16-CT	0.25	0.488
		17-CT	0.5	0.977
	MDR	18-CT	0.25	0.488
	NOT MDR	19-CT	0.25	0.488
		20-CT	0.5	0.977
*E. faecium*	VRE	21-CT	0.25	0.488
		22-CT	1	1.95
	Lin R	23-CT	0.5	0.977
		24-CT	1	1.95
	MDR	25-CT	0.12	0.234
		26-CT	1	1.95
	NOT MDR	27-CT	2	3.91

**Table 3 antibiotics-11-00767-t003:** Activity of PB4 on *E. coli* ATCC 25922 control strain and on 12 *E. coli* clinical strains (ESBL and susceptible).

Antimicrobial Resistance Profile	PB4 MIC		
	Strain	mg/L	µmol/L
	*E. coli* ATCC 25922	1	1.95
ESBL	28-CT	4	7.82
	29-CT	16	31.28
	30-CT	64	125.12
Susceptible	31-CT	4	7.82
	32-CT	4	7.82
	33-CT	4	7.82
	34-CT	4	7.82
	35-CT	4	7.82
	36-CT	8	15.64
	37-CT	32	62.56
	38-CT	32	62.56
	39-CT	64	125.12

**Table 4 antibiotics-11-00767-t004:** Activity of PB4 on *A. baumannii* ATCC 17978 control strain and 14 CRAB clinical strains.

Antimicrobial Resistance Profile	PB4 MIC		
	Strain	mg/L	µmol/L
	*A. baumannii* ATCC 17978	0.5	0.97
CRAB	40-CT	<0.12	<0.23
	41-CT	<0.12	<0.23
	42-CT	<0.12	<0.23
	43-CT	<0.12	<0.23
	44-CT	0.25	0.488
	45-CT	0.25	0.488
	46-CT	0.25	0.488
	47-CT	0.25	0.488
	48-CT	0.25	0.488
	49-CT	0.25	0.488
	50-CT	0.5	0.977
	51-CT	0.5	0.977
	52-CT	0.5	0.977
	53-CT	1	1.955

**Table 5 antibiotics-11-00767-t005:** The table reports the MIC values of PB4 on the control strains used as references. The number of clinical strains tested for each species, the PB4 MIC ranges obtained and the distribution of the results are shown alongside.

		**PB4 MIC Value (mg/L)**													
**Control Strain**	**MIC Value (mg/L)**	**Species**	**n.**	**Range**	**128**	**64**	**32**	**16**	**8**	**4**	**2**	**1**	**0.5**	**0.25**	**0.12**
*S. aureus* ATCC 29213	0.25	*S. aureus*	14	0.12–0.5	0	0	0	0	0	0	0	0	2	1	11
*E. faecalis* ATCC 29212	0.5	*E. faecalis*	6	0.25–0.5	0	0	0	0	0	0	0	0	2	4	0
*E. faecalis* ATCC 51299 VRE	0.5	*E. faecium*	7	0.12–2	0	0	0	0	0	0	1	3	1	1	1
*E. coli* ATCC 25922	1	*E. coli*	12	4–64	0	2	2	1	1	6	0	0	0	0	0
*A. baumannii* ATCC 17978	0.5	*A. baumannii*	14	≤0.12–1	0	0	0	0	0	0	0	1	3	6	4
		**PB4 MIC Value (µmol/L)**													
**Control Strain**	**MIC Value (mg/L)**	**Species**	**n.**	**Range**	**250.2**	**125.1**	**62.5**	**31.2**	**15.6**	**7.8**	**3.9**	**1.9**	**0.977**	**0.488**	**0.234**
*S. aureus* ATCC 29213	0.488	*S. aureus*	14	0.234–0.977	0	0	0	0	0	0	0	0	2	1	11
*E. faecalis* ATCC 29212	0.977	*E. faecalis*	6	0.488–0.977	0	0	0	0	0	0	0	0	2	4	0
*E. faecalis* ATCC 51299 VRE	0.977	*E. faecium*	7	0.234–3.91	0	0	0	0	0	0	1	3	1	1	1
*E. coli* ATCC 25922	1.9	*E. coli*	12	7.82–125.12	0	2	2	1	1	6	0	0	0	0	0
*A. baumannii* ATCC 17978	0.977	*A. baumannii*	14	≤0.234–1.955	0	0	0	0	0	0	0	1	3	6	4

## Data Availability

Not applicable.

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
