# Peer review of "Heteroaryl-Ethylenes as New Effective Agents for High Priority Gram-Positive and Gram-Negative Bacterial Clinical Isolates"

_antibiotics, 2022, doi:10.3390/antibiotics11060767_

Round 1
Reviewer 1 Report
The authors should consider the followings:
1. The authors should use appropriate statistical analyses to all the tests and show if there are any statistical significances. The authors should also describe the statistical pipeline in the methodology section.
2. The authors should make a figure to show the chemical structure of PB4.
3. In the bacterial strains used in this article, did the author use any molecular method, i.e. PCR test to verify the tested bacterial strains? Please list those methods in the article and the authors may supplement such results in a supplementary section.
4. As in DMSO solvent, did the authors perform any vehicle control to the test system?
5. The authors may also provide the CAS number and/or manufacturer/batch number of PB4 used.
6. The authors should state clearly the novelty of the present article in the abstracts and conclusion.
7. In Table 1 to 4, please also list the molar concentration unit (such as micromole) of PB4 used.
8. Were the MIC tests evaluated by two independent researchers?
9. Please stated the % purity of PB4.
10. The author may increase the coverage of relevant references in the article.
11. The author may add one figure to graphically illustrate the experimental workflow.
Author Response
The authors should consider the followings:
- The authors should use appropriate statistical analyses to all the tests and show if there are any statistical significances. The authors should also describe the statistical pipeline in the methodology section.
We thank the reviewer for his suggestion. Usually when you test the susceptibility of antimicrobial molecules with MIC methods, statistical analysis is not required, as the MIC values are absolute values that asses the ability to inhibit the growth of bacteria. Moreover, there are not statistical numerical factor or variables to perform statistical analysis.
Values obtained from MIC test are absolute value for each sample. Each single value cannot be correlated with the other values, the validity of the data is given by the duplicate of the biological data, for this reason we performed two independent experiments (conducted by two different researchers) and the MIC values were the same for all the samples.
- The authors should make a figure to show the chemical structure of PB4.
We thank the Reviewer for the comment; the molecular structure has been inserted as Figure 1 in the ‘Introduction’ section.
- In the bacterial strains used in this article, did the author use any molecular method, i.e. PCR test to verify the tested bacterial strains? Please list those methods in the article and the authors may supplement such results in a supplementary section.
We thank the Reviewer for this comment, we add some sentences in the mat and methods section.
- As in DMSO solvent, did the authors perform any vehicle control to the test system?
We used DMSO at a final concentration of 1.5%. CLSI suggests, for antibiotics dissolution, not to exceed a final DMSO concentration of 1%. However, numerous publications have shown that even higher concentrations are tolerated by many microbial species. (https://doi.org/10.1002/jps.2600580708; 10.3390/mol2net-03-xxxx). Other experiments stated that for some strains like Escherichia coli ATCC 25922, Staphylococcus aureus ATCC 25923, Pseudomonas aeruginosa, ATCC 27853, Proteus mirabilis ATCC 25922 and Enterococcus faecalis ATCC 29212, DMSO is safe and innocuous at concentrations not greater than 80%. (DOI:10.5580/b43). Furthermore, a great number of studies evidenced how DMSO and ethanol are generally considered safe below 3% v/v3- However this cannot be stated as general assumption for all sample (doi: 10.3390/toxics9050092).
- The authors may also provide the CAS number and/or manufacturer/batch number of PB4 used.
We thank the Reviewer for the comment; a new paragraph has been inserted in the ‘Materials and methods’ section and the information about CAS number were added.
- The authors should state clearly the novelty of the present article in the abstracts and conclusion.
We thank to the reviewer to this comment, we added some periods to state what you suggested in abstract and discussion section.
- In Table 1 to 4, please also list the molar concentration unit (such as micromole) of PB4 used.
We thank the reviewer for this comment, we add a column in table 1 to 4, highlighted in yellow.
- Were the MIC tests evaluated by two independent researchers?
We thank to the reviewer to the comment. Yes, the MICs were performed by two different independent researchers in different weeks, in particular from AM, DB. We add a period in the text, in mat and methods section.
- Please stated the % purity of PB4.
We thank the Reviewer for the comment; the data has been inserted in the ‘Materials and methods’ section.
- The author may increase the coverage of relevant references in the article.
We thank the Reviewer for the comment, now we report 34 references vs 18 of the original article.
- The author may add one figure to graphically illustrate the experimental workflow.
We thank the Reviewer for the comment, we add a figure in the text, figure 2.
Reviewer 2 Report
This manuscript “Heteroaryl-ethylenes as new effective agents for high priority Gram-positive and Gram-negative bacterial clinical isolates” prepared by Bivona et al. presented the novelty of an antimicrobial agent PB4’s impact on clinically relevant strains. The topic of the manuscript is of great interest to nosocomial pathogen treatments, and revisions clarifying the importance of the finding will be needed.
Major:
- More information on PB4 should be introduced at the beginning of the manuscript. How has PB4 been identified, the mechanism it inhibits bacteria growth, and is it bacteria membrane targeted?
- It would be easier for readers to understand why the reference strains are selected and used in this manuscript if more information was given. Are those reference strains non-clinical strains?
- For clinical strains tested, there is increased credibility with multiple strains from the same group were used for the PB4 susceptibility test. I am just wondering for strains under the same group, are they similar in toxicity, clinical symptom or treatment response? For those strains, how similar they are when compared to their genomes. For example, in table 1, 12-14CT have higher MIC compared to the other strains in the same group, what can be the reason leading to the difference?
- In table 1, I am wondering if ATCC 29213 is an MSSA strain, (as shown in reference 17). If so, why MIC is 0.25 in ATCC 29213, compared to 0.12 for other MSSA strains. What would the MIC be like for other reference S. aureus strains?
- In table 2, E. faecalis MDR group only contains 1 strain tested for MIC, more than 2 strains in each group would be appreciated for PB4 antimicrobial effects’ prevalence among different groups.
- I noticed that E. coli has a quite big increase in MIC compared to the other species, what can be the possible explanation?
- In discussion, speculations on PB4 bacterial inhibition mechanisms and different MIC among different species or same species in different groups can help enhance the significance of the manuscript.
Minor:
- The bacteria species name should be italicized, please double-check through the manuscript.
- Some of the sentences are redundant and complicated, which leads to confusion.
- Line 26: “standard method, with some modifications”, is not very accurate in describing the MIC method in the abstract.
- Line 141-144: Breaking up the long sentence into several short sentences can be helpful.
- Line 173: On the left are the reported MIC values ….
- The MIC method should be further illustrated, e.g.: how does the MIC concentration determined? Is it by naked eye observation of no growth in the well, OD600 test or accompanied with a growth curve validation?
Author Response
This manuscript “Heteroaryl-ethylenes as new effective agents for high priority Gram-positive and Gram-negative bacterial clinical isolates” prepared by Bivona et al. presented the novelty of an antimicrobial agent PB4’s impact on clinically relevant strains. The topic of the manuscript is of great interest to nosocomial pathogen treatments, and revisions clarifying the importance of the finding will be needed.
Major:
- More information on PB4 should be introduced at the beginning of the manuscript. How has PB4 been identified, the mechanism it inhibits bacteria growth, and is it bacteria membrane targeted?
We thank the Reviewer for the comment; the Introduction section has been improved following reviewer suggestion and a new paragraph has been inserted in the ‘Materials and methods’ section.
The PB4 mechanism of action, currently under study, is out of the scope of this manuscript focused on the deep investigation of his inhibition efficacy on bacterial clinical isolates.
- It would be easier for readers to understand why the reference strains are selected and used in this manuscript if more information was given. Are those reference strains non-clinical strains?
We thank to the reviewer; we use the international reference strains (ATCC) that are included in the list of reference strains supported by EUCAST and CLSI (CLSI. 2020. CLSI supplement M100. 30th edition. Wayne: Clinical and Laboratory Standards Institute). In particular we selected a reference strain for each species as recommended by CLSI. The MICs for PB4 compound were reported in our first paper, in which we evaluate our compound’s activity spectrum and antibacterial ability, minimum inhibitory concentration (MIC) and minimum bactericidal concentration (MBC) tests performed on Gram-positive and Gram-negative ATCC strains.
- For clinical strains tested, there is increased credibility with multiple strains from the same group were used for the PB4 susceptibility test. I am just wondering for strains under the same group, are they similar in toxicity, clinical symptom or treatment response? For those strains, how similar they are when compared to their genomes. For example, in table 1, 12-14CT have higher MIC compared to the other strains in the same group, what can be the reason leading to the difference?
We thank to the reviewer for this comment. We tried to select our clinical isolates in the most various way possible, without taking into account the characteristics strictly related to the patient; the only difference we made was the distinction by antibiotic profile. In particular MSSA and MRSA for S. aureus; VRE, MDR and non-MDR for E. faecalis; VRE, LINR MDR and non-MDR for E. faecium, ESBL or sensitive for E. coli and CRAB for A. baumanii.
Our aim was to understand if there could actually be differences in MICs values within the resistant/susceptible subpopulation. Although the values can be different between the isolates, 13CT and 14CT (0.5mg/L) and the rest of the staphs, in reality have only a MIC value above the ATCC29123 (0.25mg/L), proving to be quite homogeneous.
The same is true for the group of enterococci and for Acinetobacter, which indeed show lower MICs values than the reference strain. As we have discussed in the text, the case of E. coli is different.
- In table 1, I am wondering if ATCC 29213 is an MSSA strain, (as shown in reference 17). If so, why MIC is 0.25 in ATCC 29213, compared to 0.12 for other MSSA strains. What would the MIC be like for other reference S. aureus strains?
As regards clinical strains, experiments have been repeated in duplicate to obtained more reliable MIC values. In addition, MIC values on ATCC control strains had already been obtained in our previous study (D. Bongiorno et al., “Heteroaryl-Ethylenes as New Lead Compounds in the Fight against High Priority Bacterial Strains.,” Antibiotics (Basel, Switzerland), vol. 10, no. 9, Aug. 2021, doi: 10.3390/antibiotics10091034); in this study we confirmed previous results.
ATCC29213 is the standard strains for S. aureus. Concerning MSSA PB4 MIC values (0,25 vs 0,12 μg/ml), we do not consider significant the dilution difference (only one dilution), since all MSSA strains resulted susceptible to PB4. Moreover, the same condition has been highlighted for other bacteria such as E. coli and A. baumannii.
- In table 2, E. faecalis MDR group only contains 1 strain tested for MIC, more than 2 strains in each group would be appreciated for PB4 antimicrobial effects’ prevalence among different groups.
Thank you for your precision. Unfortunately, at that moment we had only one MDR E. faecalis strain to test with PB4. Furthermore, even with only one MDR strain, we did not find significant differences in terms of PB4 susceptibility when compared to non-MDR strains.
- I noticed that E. coli has a quite big increase in MIC compared to the other species, what can be the possible explanation?
Thank you for your question. PB4 is a new compound with unknown PK/PD characteristics. This is the first study in which PB4 has been tested on clinical strains and every result we found will be further analyze and tested again in other studies to better explain its antibacterial mechanisms and pharmacological features. We stated that within the discussion section.
- In discussion, speculations on PB4 bacterial inhibition mechanisms and different MIC among different species or same species in different groups can help enhance the significance of the manuscript.
Thank you for your observation. We have been implemented discussion section with different consideration on PB4 action.
Minor:
- The bacteria species name should be italicized, please double-check through the manuscript.
Done all over the text.
- Some of the sentences are redundant and complicated, which leads to confusion.
Done all over the text.
- Line 26: “standard method, with some modifications”, is not very accurate in describing the MIC method in the abstract.
The modifications made to the standard methods were reported in Mat and Methods section, since in the abstract we cannot exceeded to 250 words, we change the sentence in the abstract.
- Line 141-144: Breaking up the long sentence into several short sentences can be helpful.
We selected 14 clinical MDR strains: particular attention was paid to their carbapenem-resistance profiles - Carbapenem Resistant A. baumannii (CRAB). We obtained PB4 MIC values comprised in a range between <0,12 mg/L and 1 mg/L (see Table 4).
Changed as suggested.
- Line 173: On the left are the reported MIC values ….
We change the table.
- The MIC method should be further illustrated, e.g.: how does the MIC concentration determined? Is it by naked eye observation of no growth in the well, OD600 test or accompanied with a growth curve validation?
The MIC concentration was evaluated as reported in the standard method, CLSI 2020 and EUCAST 2022 (CLSI supplement M100. 30th edition. Wayne: Clinical and Laboratory Standards Institute and EUCAST; https://www.eucast.org/fileadmin/src/media/PDFs/EUCAST_files/Disk_test_documents/2022_manuals/Reading_guide_BMD_v_4.0_2022.pdf) without any further experiment cause we asses previously MIC and MBC values for all the ATCC strains used in this paper. We add 3 ref in the mat and method section.
Round 2
Reviewer 2 Report
Thank you for the reply, please check figure 2 display, it seems overlapped with the other picture.